# HR-TD: A Regularized TD Method to Avoid Over-Generalization

## Abstract

Temporal Difference learning with function approximation has been widely used recently and has led to several successful results. However, compared with the original tabular-based methods, one major drawback of temporal difference learning with neural networks and other function approximators is that they tend to over-generalize across temporally successive states, resulting in slow convergence and even instability. In this work, we propose a novel TD learning method, **H**adamard product **R**egularized **TD** (HR-TD), to reduce over-generalization and thus leads to faster convergence. This approach can be easily applied to both linear and nonlinear function approximators. HR-TD is evaluated on several linear and nonlinear benchmark domains, where we show improvement in learning behavior and performance.

## 1 Introduction

Temporal Difference (TD) learning is one of the most important paradigms in Reinforcement Learning (Sutton and Barto, 1998). Techniques based on combining TD learning with nonlinear function approximators and stochastic gradient descent, such as deep networks, have led to significant breakthroughs in large-scale problems to which these methods can be applied (Mnih et al., 2015; Silver et al., 2016; Schulman et al., 2015).

At its heart, the TD learning update is straightforward. $v(s)$ estimates the value of being in a state $s$. After an action $a$ that transitions the agent from $s$ to next state $s'$, $v(s)$ is altered to be closer to the (discounted) estimated value of $s'$, $v(s')$ (plus any received reward, $r$). The difference between these estimated values is called the temporal difference error (TD error) and is typically denoted as $\delta$. Formally, $\delta = r + \gamma v(s') - v(s)$, where $\gamma$ is the discount factor, and $r + \gamma v(s')$ is known as the TD target.

When states are represented individually (the *tabular* case), $v(s)$ can be altered independently from $v(s')$ using the update rule $v(s) \leftarrow v(s) + \alpha\delta$, where $\alpha$ is the learning rate. In fully deterministic environments, $\alpha$ can be set to 1, thus causing $v(s)$ to change all the way to the TD target. Otherwise, in a stochastic environment, $\alpha$ is set less than 1 so that $v(s)$ only moves part of the way towards the TD target, thus avoiding over-generalization from a single example. When, on the other hand, states are represented with a function approximator, as is necessary in large or continuous environments, $v(s)$ can no longer be updated independently from $v(s')$. That is because $s$ and $s'$ are likely to be similar (assuming actions have local effects), any change to $v(s)$ is likely to also alter $v(s')$. While such generalization is desirable in principle, it also has the unintended consequence of changing the TD target, which in turn can cause the TD update to lead to an increase in the TD error between $s$ and $s'$. This unintended consequence can be seen as a second form of over-generalization: one that can be much more difficult to avoid.

Past work has identified this form of over-generalization in RL, has observed that it is particularly relevant in methods that use neural network function approximators such as DQN (Mnih et al., 2015), and has proposed initial solutions (Durugkar and Stone, 2017; Pohlen et al., 2018). In this paper, we present a deeper analysis of the reasons for this form of over-generalization and introduce a novel learning algorithm termed HR-TD, based on the recursive proximal mapping formulation of TD learning (Bertsekas, 2011), which offers a mathematical framework for parameter regularization that allows one to control for this form of over-generalization. Empirical results across multiple

domains demonstrate that our novel algorithm learns more efficiently (from fewer samples) than prior approaches.

The rest of the paper is organized as follows. Section 2 offers a brief background on TD learning, the over-generalization problem, and optimization techniques used in the derivation of our algorithm. In Section 3, we discuss the state-of-the-art research in this direction. The motivation and the design of our algorithm are presented in Section 4. Finally, the experimental results of Section 5 validate the effectiveness of the proposed algorithm.

## 2 BACKGROUND

This section builds on the notation introduced in Section 1 to specify the problem of interest in full detail. We introduce the background for TD learning, over-generalization, and proximal mapping, which are instrumental in the problem formulation and algorithm design.

### 2.1 REINFORCEMENT LEARNING AND OVER-GENERALIZATION

Reinforcement Learning problems are generally defined as Markov Decision Processes (MDPs). We use the definition and notation as used in Sutton and Barto (2017), unless otherwise specified. In this paper, we focus on domains with large or continuous state spaces such that function approximation is needed. We define the value estimate of state $s$ with parameter $\theta$ when following policy $\pi$ as, $v_\pi(s|\theta) = \mathbb{E}_\pi \left[ R_t + \gamma R_{t+1} + \gamma^2 R_{t+2} + \ldots | S_t = s \right]$. Here $R_t$ is the random variable associated with a reward at time $t$, and $r_t$ is used as an instantiation of this random variable. The optimal (true) value function $v_\pi^*$ satisfies the Bellman equation given as $v_\pi^*(s|\theta) = \mathbb{E}_\pi \left[ R_t + \gamma v_\pi^*(s'|\theta) \right]$. During TD learning, the estimated value function is altered to try to make this property hold. In effect, state values are updated by bootstrapping off of the estimated value of the predicted next states.

We focus on 1-step TD methods, i.e., TD(0), that bootstrap from the value of the immediate next state or states in the MDP to learn the value of the current state. The TD error $\delta_t(s_t, s_{t+1}|\theta)$ to be minimized is as follows:

$$\delta_t(s_t, s_{t+1}|\theta) = (r_t + \gamma v_\pi(s_{t+1}|\theta)) - v_\pi(s_t|\theta)$$

In the following, $\delta_t(s_t, s_{t+1}|\theta)$ is written as $\delta_t$ for short. When using function approximation and gradient descent to optimize the parameters, the loss to be minimized is the squared TD error. At the $t$-th time-step, the objective function used in TD learning is $\mathcal{L}_{TD} = \|r_t + \gamma v_\pi(s_{t+1}|\theta) - v_\pi(s_t|\theta)\|^2$.

Similarly, the optimal action value function $Q$ satisfies the Bellman optimality equation $Q^*(s_t, a_t|\theta) = R_t + \gamma \max_a Q^*(s_{t+1}, a|\theta)$. The objective used in Q-Learning is thus $\mathcal{L}_Q = \|r_t + \gamma \max_a Q(s_{t+1}, a|\theta) - Q(s_t, a_t|\theta)\|^2$.

The partial derivative of $v(s_t|\theta)$ or $Q(s_t, a_t|\theta)$ with respect to $\theta$ is the direction in which TD learning methods update the parameters. We use $g_t(s_t|\theta)$ and $g_t(s_t, a_t|\theta)$ to refer to these vectors. In the linear case, $v(s_t|\theta) = \theta_t^\top \phi(s_t)$, where $\phi(s_t)$ are the features of state $s_t$. In this case, $g_t(s_t, a_t|\theta)$ is the feature vector $\phi(s_t, a_t)$, and in general, $g_t(s_t, a_t|\theta) = \partial_\theta Q(s_t, a_t|\theta)$. It is computed as:

$$g_t(s_t, a_t|\theta) = \frac{\partial Q(s_t, a_t|\theta)}{\partial \theta}$$
$$\theta \leftarrow \theta + \alpha \delta_t g_t(s_t, a_t|\theta).$$

We have already briefly alluded to the issue of over-generalization in Section 1. One of the reasons we use function approximation is that we want the values we learn to generalize to similar states. But one of these similar states is likely to be the target of our Bellman equation $v(s_{t+1}|\theta)$. If the weights that correspond to large or important features in $\phi(s_{t+1})$ are strengthened, then the TD error might not decrease as much as it should, or it might even increase. We refer to parameter updates that work against the objective of reducing the TD error as over-generalization.

### 2.2 PROXIMAL MAPPING FORMULATION OF TD LEARNING

In this section, we introduce the basics of proximal mapping, which provide the mathematical formulation of our algorithm design. A proximal mapping (Parikh and Boyd, 2013) $\text{prox}_f(w)$ associated

with a convex function $f$ is defined as

$$\text{prox}_f(w) = \arg\min_x \left( f(x) + \frac{1}{2} \|w - x\|_2^2 \right) \tag{1}$$

Such a proximal mapping is typically used after a parameter update step to incorporate constraints on the parameters. Intuitively, the first term $f(x)$ provides incentive to move $x$ in the direction that minimizes $f$, whereas the second term $\frac{1}{2}\|w - x\|_2^2$ provides pressure to keep $x$ close to $w$. If $f(x) = 0$, then $\text{prox}_f(w) = w$, the identity function. $f$ can often be a regularization term to help incorporate prior knowledge. For example, for learning sparse representations, the case of $f(x) = \beta\|x\|_1$ is particularly important. In this case, the entry-wise proximal operator is:

$$\text{prox}_f(w)_i = \text{sign}(w_i)\max(|w_i| - \beta, 0)$$

Proximal methods have been shown to be useful for various reinforcement learning problems, e.g., proximal gradient TD learning (Liu et al., 2015) integrates the proximal method with gradient TD learning (Sutton et al., 2009) using the Legendre-Fenchel convex conjugate function (Boyd and Vandenberghe, 2004), and projected natural actor-critic (Thomas et al., 2013) interprets natural gradient as a special case of proximal mapping. We now introduce the *recursive proximal mapping* formulation of TD learning algorithm (Bertsekas, 2011). At the $t$-th iteration, the TD update law solves a *recursive proximal mapping*, i.e., $\theta_{t+1} = \theta_t + \alpha_t \delta_t g_t(s_t)$, which is equivalent to

$$\theta_{t+1} = \arg\min_x \left\{ \langle x, -\delta_t g_t(s_t) \rangle + \frac{1}{2\alpha_t} \|x - \theta_t\|_2^2 \right\} \tag{2}$$

It should be noted that Eq. (2) is different from Eq. (1) in that Eq. (1) has an explicit objective function $f$ to optimize. Eq. (2) does not have an explicit objective function, but rather corresponds to a fixed-point equation. In fact, it has been proven that the TD update term $\delta_t g_t(s_t)$ does not optimize any objective function (Maei, 2011). Discussing this in details goes beyond the scope of the paper, and we refer interested readers to (Maei, 2011; Bertsekas, 2011) for a comprehensive discussion of this topic.

## 3 RELATED WORK

To the best of our knowledge, the closest work to ours to address the over-generalization problem is the Temporal Consistency loss (TC-loss) method (Pohlen et al., 2018) and the constrained TD approach (Durugkar and Stone, 2017).

The TC-loss (Pohlen et al., 2018) aims to minimize the change to the target state by minimizing explicitly a separate loss that measures change in the value of $s'$, i.e., $\mathcal{L}\big(V_\theta(s', a') - V_{\theta_{t-1}}(s', a')\big)$. When used in conjunction with a TD loss, it guarantees that the updated estimates adhere to the Bellman operator and thus are temporally consistent. However, there are some drawbacks to this method. Firstly, the asymptotic solution of the TC-loss method is different from the TD solution due to the two separate losses, and the solution property remains unclear. Secondly, each parameter component plays a different role in changing $v(s')$. For instance, if the component of $\theta$ is or close to 0, then this component does not have any impact on changing $v(s')$. Different parameter components, therefore, should be treated differently according to their impact on the value function changes (or action-value change in case of DQN).

Another recent work in this direction is the constrained TD (CTD) algorithm (Durugkar and Stone, 2017). To avoid the over-generalization among similar sates, CTD tends to alleviate over-generalization by using the vector rejection technique to diminish the update along the direction of the gradient of the action-value function of the successive state. In other words, the real update is made to be orthogonal to the gradient of the next state. However, the CTD method suffers from the double-sampling problem, which is explained in detail in Appendix A. Moreover, since it mainly uses vector rejection, this method is not straightforward to extend to nonlinear function approximation, such as the DQN network, where over-generalization can be severe. Lastly, if the state representation of $s_t$ and $s_{t+1}$ are highly similar, as in case of visual environments like Atari games, then the vector rejection causes the update to be almost orthogonal to the computed gradient.

## 4 HADAMARD PRODUCT REGULARIZED TD

In this section, we analyze the reason for over-generalization and propose a novel algorithm to mitigate it.

### 4.1 ANALYSIS OF OVER-GENERALIZATION

Consider the update to the parameter $\theta_t$ as follows, with TD error $\delta_t$, learning rate $\alpha$ and a linear function approximation $v(s_t|\theta_t)$ with features $\phi(s_t)$ and gradient $g(s_t|\theta_t) = \phi(s_t)$.

$$\theta_{t+1} = \theta_t + \alpha\delta(s_t, s_{t+1}|\theta_t)\phi(s_t)$$

If we substitute the above value for $\theta_{t+1}$, the TD error for the same transition after the update is

$$\begin{aligned}
\delta(s_t, s_{t+1}|\theta_{t+1}) &= r_t - (\theta_{t+1}^\top\phi(s_t) - \gamma\theta_{t+1}^\top\phi(s_{t+1})) \\
&= \delta(s_t, s_{t+1}|\theta_t) - \alpha\delta(s_t, s_{t+1}|\theta_t)\left(\phi(s_t)^\top\phi(s_t) - \gamma\phi(s_t)^\top\phi(s_{t+1})\right),
\end{aligned}$$

and thus

$$\delta(s_t, s_{t+1}|\theta_t) - \delta(s_t, s_{t+1}|\theta_{t+1}) = \alpha\delta(s_t, s_{t+1}|\theta_t)\left(\phi(s_t)^\top\phi(s_t) - \gamma\phi(s_t)^\top\phi(s_{t+1})\right).$$

We see above that the decrease in the TD error at $t$ depends on two factors, the inner product of the gradient with features of $s_t$, and its inner product with the features of $s_{t+1}$. This decrease will be reduced if $\phi(s_t)$ and $\phi(s_{t+1})$ have a large inner product. If this inner product exceeds $\frac{1}{\gamma}\phi(s_t)^\top\phi(s_t)$, then in fact the error increases. Thus over-generalization is an effect of a large positive correlation between the update and the features of $s_{t+1}$, especially when contrasted with the correlation of this same update with the features of $s_t$.

We are then left with the following question: what kind of weight update can maximize the reduction in $\delta_t$? Merely minimizing the correlation of the update with $\phi(s_{t+1})$ is insufficient, as it might lead to minimizing the correlation with $\phi(s_t)$. This is the issue that Constrained TD (Durugkar and Stone, 2017) faces with its gradient projection approach. Hence, we must also maximize its correlation with $\phi(s_t)$.

To examine this effect, we consider the properties of parameters that we should avoid changing, to the extent possible. Consider the linear value function approximation case: $v_\theta(s) = \phi(s)^\top\theta$. For example, consider $s_t$ and $s_{t+1}$ with the features $\phi(s_t) = [0, 2, 1]$, and $\phi(s_{t+1}) = [2, 0, 1]$. Then for two different weights, $\theta_1 = [0, 0, 2]$ and $\theta_2 = [1, 1, 0]$, we have the same value for both these parameter vectors at both $s_t$ and $s_{t+1}$, i.e. $\phi(s_t)^\top\theta_1 = \phi(s_{t+1})^\top\theta_1 = \phi(s_t)^\top\theta_2 = \phi(s_{t+1})^\top\theta_2 = 2$. However, the results of the Hadamard product ($\circ$) of these parameters with the feature vectors are different, i.e.

$$\phi(s_t) \circ \theta_1 = \phi(s_{t+1}) \circ \theta_1 = [0, 0, 2],$$
$$\phi(s_t) \circ \theta_2 = [0, 2, 0], \quad \phi(s_{t+1}) \circ \theta_2 = [2, 0, 0],$$

where the Hadamard products of $\theta_1$ with $\phi(s_t)$ and $\phi(s_{t+1})$ are more correlated than those of $\theta_2$. An update to the last weight of $\theta_1$ will cause the values of both $s_t$ and $s_{t+1}$ to change, but an update to the second weight of $\theta_2$ will affect only $s_t$. In fact, unless both the first and the second weights change, $s_t$ and $s_{t+1}$ do not change simultaneously. In this sense, $\theta_1$ tends to cause aggressive generalization across the values of $s_t$ and $s_{t+1}$, and thus the TD update to $\theta_1$ should be regularized more heavily. The Hadamard product of the weights and the features allows us to distinguish between $\theta_1$ and $\theta_2$ in this way.

Motivated by this observation, we aim to reduce the over-generalization by controlling the weighted feature correlation between the current state $g(s)\circ\theta$ and the successive state $g(s')\circ\theta$, i.e., $\text{Corr}(g(s)\circ\theta, g(s')\circ\theta)$.

### 4.2 ALGORITHM DESIGN

Given the constraint as shown above, the constrained Mean-Squares Error (MSE) is formulated as

$$\theta^* = \arg\min_\theta \frac{1}{2}||V - v_\theta||_2^2, \quad \text{s.t. } \text{Corr}(g(s) \circ \theta, g(s') \circ \theta) \leq \rho, \tag{3}$$

---

**Algorithm 1** Hadamard product Regularized TD (HR-TD) Learning

---

**Require:** $T$, $\alpha_t$(learning rate), $\gamma$(discount factor), $\eta$(initial regularization parameter).
**Ensure:** Initialize $\theta_0$.
   **for** $t = 1, 2, 3, \cdots, T$ **do**
      $\eta_t = \eta/t$
      Update $\theta_{t+1}$ according to Eq. (5).
   **end for**

---

where $V$ is the true value function. Using the *recursive proximal mapping* with respect to the constrained objective function, the parameter update per-step of Eq. (3) can be written as

$$\theta_{t+1} = \arg\min_\theta \big\{ - \theta^\top \big(\mathbb{E}[\delta_t] g(s_t)\big) + \frac{1}{2\alpha_t} ||\theta - \theta_t||_2^2 \big\}, \quad \text{s.t. } \mathrm{Corr}(g(s_t) \circ \theta, \; g(s_{t+1}) \circ \theta_t) \leq \rho.$$

Using Lagrangian duality, it can be reformulated as

$$\theta_{t+1} = \arg\min_\theta \big\{ - \theta^\top \big(\mathbb{E}[\delta_t] g(s_t)\big) + \frac{1}{2\alpha_t} ||\theta - \theta_t||_2^2 + \eta \mathrm{Corr}(g(s_t) \circ \theta, \; g(s_{t+1}) \circ \theta_t) \big\},$$

where $\eta$ is the factor that weights the constraint against the objective. The closed-form solution to the weight update is

$$\theta_{t+1} = \theta_t + \alpha_t \big(\mathbb{E}[\delta_t] g(s_t) - \eta(g(s_t) \circ g(s_{t+1}) \circ \theta_t)\big) \tag{4}$$

Using sample-based estimation, i.e., using $g_t(s)$ (resp. $g_t(s')$) to estimate $g(s)$ (resp. $g(s')$), and using $\delta_t$ to estimate $\mathbb{E}[\delta_t]$, the Eq. (4) becomes

$$\theta_{t+1} = \theta_t + \alpha_t \big(\delta_t g_t(s_t) - \eta(g_t(s_t) \circ g_t(s_{t+1}) \circ \theta_t)\big) \tag{5}$$

In the proposed algorithm, if the component of the weights helps decrease the Hadamard product correlation, then it is not penalized. Now the algorithm for value function approximation is formulated as in Algorithm 1, and the algorithm for control is formulated in Algorithm 2.

### 4.3 Hadamard product Regularized Deep Q Network

In DQN, the value function is learned by minimizing the following squared Bellman error using SGD and backpropagating the gradients through the parameter $\theta$

$$\mathcal{L}_{\text{DQN}} = \frac{1}{2} ||r_t + \gamma Q(s_{t+1}, a_{t+1}|\theta') - Q(s_t, a_t|\theta)||^2. \tag{6}$$

Here, $\theta'$ are the parameter of the target network that is periodically updated to match the parameters being trained. The action $a_{t+1}$ is chosen as $\arg\max_a Q(s_{t+1}, a|\theta')$ if we use DQN, and $\arg\max_a Q(s_{t+1}, a|\theta)$ if we use Double DQN (DDQN) (Van Hasselt et al., 2016). We use DDQN in experiments as DQN has been shown to over-estimate the target value.

Let $\phi(s_t|\theta)$ be the activations of the last hidden layer before the Q-value calculation and $\theta_{-1}$ be the corresponding weights for this layer. The Correlation can be written as $\mathcal{L}_{\text{corr}} = \mathrm{Corr}(\phi(s_t|\theta) \circ \theta, \; \phi(s_{t+1}|\theta) \circ \theta_t)$. We do not use the target network when calculating this loss. The loss used in Hadamard regularized DDQN is then an $\eta$-weighted mixture of Eq. (6) and this loss

$$\mathcal{L}_{\text{HR-TD}} = \mathcal{L}_{\text{DQN}} + \eta \mathcal{L}_{\text{corr}} \tag{7}$$

### 4.4 Theoretical Analysis

In this section, we conduct some initial analysis of Algorithm 1 with linear function approximation. For simplicity, we only discuss the linear case, i.e., $\partial v_\theta(s_t) = \phi(s_t)$, $\partial v_\theta(s_{t+1}) = \phi(s_{t+1})$. If Algorithm 1 converges, the update of the weights according to Eq. (5) should satisfy the following condition

$$\mathbb{E}[\delta_t \phi(s_t) - \eta(\phi(s_{t+1}) \circ \theta_t \circ \phi(s_t))] = 0.$$

| Approximator | TD | TD+TC | HR-TD |
|---|---|---|---|
| Fourier Basis | **691.99** | 736.29 | **691.93** |
| MLP | 239.73 | 716.546 | **232.831** |

Table 1: Mean Square Error (MSE) averaged across 10 runs

Rewriting $\delta_t$ and denoting $\Delta\phi_t = \phi(s_t) - \gamma\phi(s_{t+1})$, we have

$$\mathbb{E}[\phi(s_t)r_t] = \mathbb{E}[\phi(s_t)\Delta\phi_t^\top + \eta M]\theta_t,$$

where $M = \mathbb{E}[\text{Diag}(\phi(s_t) \circ \phi(s_{t+1}))] = \mathbb{E}[\text{Diag}(\phi(s_t)\phi^\top(s_{t+1}))]$. Thus we have

$$\mathbb{E}[\phi(s_t)(\Delta\phi(s_t))^\top + \eta M] = \mathbb{E}[\phi(s_t)\phi^\top(s_t) - \gamma\phi(s_t)\phi^\top(s_{t+1}) + \eta\text{Diag}(\phi(s_t)\phi^\top(s_{t+1}))]$$

If we set $\eta \to \gamma$, we observe that the second and third terms in the RHS above cancel out in the diagonal element. Consider the scheme where we initialize $\eta = \gamma$ and then reduce it as over the training process. It is equivalent to slowly introducing the discount factor into the error computation. It has been shown (Prokhorov and Wunsch, 1997) that instead of the discount factor $\gamma$ provided by the MDP, a user-defined time-varying $\gamma_t$ can help accelerate the learning process of the original MDP w.r.t $\gamma$. This previous work suggests using a small discount factor $\gamma_t < \gamma$ in the beginning, and then increasing $\gamma_t$ gradually to $\gamma$. HR-TD results in a similar effect without defining a separate $\gamma_t$ and its schedule.

## 5 EXPERIMENTS

We evaluate HR-TD on two classical control problems: Mountain Car and Acrobot using both linear function approximation with Fourier basis features and nonlinear function approximation using Deep Neural Networks. We verify that this algorithm scales to complex domains such as the Atari Learning Environment (Bellemare et al., 2013), by evaluating our approach on the game of Pong. We utilize OpenAI gym (Brockman et al., 2016) to interface our agent with the environments. We compare HR-TD to the baselines by using the following metrics: **1)** Accumulated reward per episode. **2)** Average change in the target Q value at $s'$ after every parameter update. For comparison, we consider Q learning and Q learning with TC loss (and DDQN for neural networks).

Based on our analysis, we expect HR-Q learning to begin improving the policy earlier in the learning process, and we expect HR-TD to be able to evaluate the value of a policy just as well as TD. We evaluate the change of the value of the next state as well, and consider whether HR-TD is able to reduce this change as a consequence of the regularization. We note, however, that this quantity is diagnostic in nature, rather than being the true objective. It would definitely be possible to minimize this quantity by making no learning updates whatsoever, but then we would also observe no learning.

### 5.1 EVALUATION

Before we consider the effect of HR-Q on control tasks, we compare the purely evaluative property of HR-TD. Here, we evaluate a trained policy on the Mountain Car domain. We run this experiment

---

**Algorithm 2** Hadamard product Regularized Q (HR-Q) Network

**Require:** $T$, $\alpha_t$(learning rate), $\gamma$(discount factor), $\eta$(initial regularization parameter).
**Ensure:** Initialize $\theta_0$.
  **repeat**
    $\eta_t = \eta/t$
    Choose $a_t$ using policy derived from $Q$ (e.g., $\epsilon$-greedy)
    Take $a_t$, observe $r_t, s_{t+1}$
    Add $s_t, a_t, r_t, s_{t+1}$ to Replay Buffer
    Sample batch from Buffer and Update $\theta_{t+1}$ using backpropagation to minimize Eq. (7).
    $t \leftarrow t + 1$
  **until** training done

---

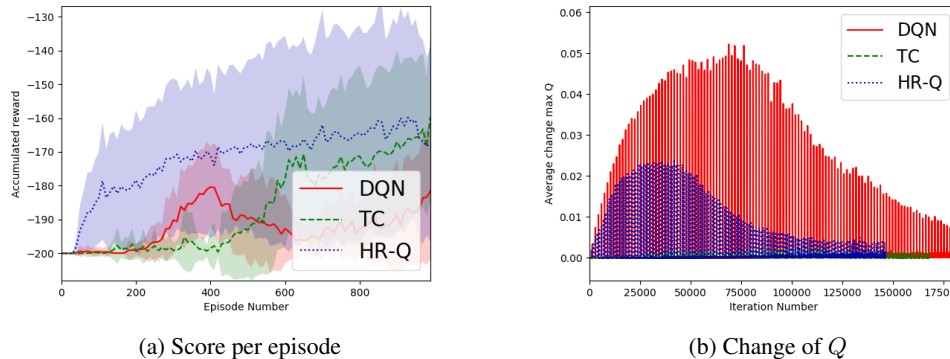

(a) Score per episode            (b) Change of $Q$

Figure 1: Comparison of performance of HR-Q with a neural network on the Mountain Car domain. Figure 1a shows the cumulative score in an episode on the y-axis, with the episode number depicted on the x-axis. 1b compares how much the value of the TD target changed after an update. The x-axis is number of iterations

10 times for each method. For each experiment, the policy is executed in the environment for 10000 steps, resetting the agent to one of the start states if it terminates. We learn the value function using TD by sampling a batch of transitions from this dataset and take 10,000 learning steps per run.

The metric we compare is the MSE with the Monte Carlo estimate of the same run, taken over 300,000 transitions. The MSE value for each experiment is calculated by averaging the MSE of the last 10 training steps, to reduce sampling error. Finally, we take the mean of these errors across the 10 runs and present them in Table 1. TD and HR-TD reach roughly the same value for all the runs. TC, however, converges to a different minimum that leads to a very high MSE. This may be because the competing TD and TC objectives in this method cause the learning to destabilize. If we lower the learning rate for TC, then we avoid this behavior but also do not converge in the given max number of training steps.

## 5.2 NEURAL NETWORKS

We now consider the performance of HR-Q learning when using Neural Networks for function approximation. We consider two domains, Mountain Car and Acrobot, but we do not perform any basis expansion and feed the state values directly into a neural network with a single hidden layer of 64 units.

We compare the performance of HR-Q in Figure 1 and 2, with Q-Learning and Q-learning with TC loss. We use DDQN Van Hasselt et al. (2016) as the underlying algorithm for Q-learning. Details of

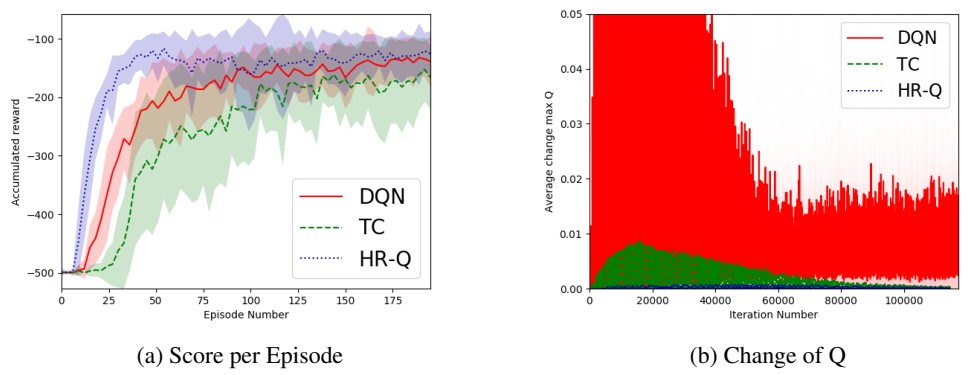

(a) Score per Episode            (b) Change of Q

Figure 2: Comparison of performance of HR-Q with a neural network on the Acrobot domain

the network and hyperparameters are in Appendix B. We take 20 independent runs, with a different seed in each run used to initialize Tensorflow, NumPy, and the OpenAI Gym environment. Each run is taken over 1000 episodes. In both these experiments, we see HR-TD starts to learn a useful policy behavior before either of the other techniques. Interesting to note is that in Fig. 1b, HR-TD learns a state representation that causes the target value to change less than DQN but does not restrict it as much as TC. But in Fig. 2b we see that HR-TD is able to find a representation that is better at keeping the target value separate than TC is. However, in both these cases, the value function that is learned seems to be quickly useful for learning a better policy.

## 5.3 ATARI

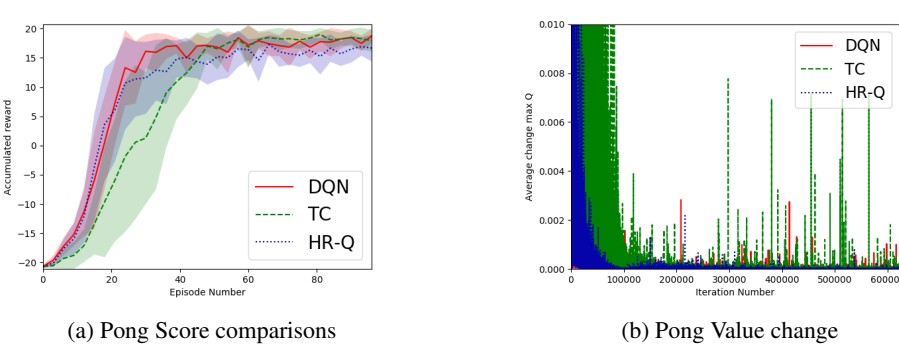

(a) Pong Score comparisons       (b) Pong Value change

Figure 3: Experimental Evaluation on Atari Pong domain

We also validate the applicability of this technique to a more complex domain and a more complex network. We apply the HR-Q to DDQN on the Atari domain to verify that the technique is scalable and that the findings and trends we see in the first two experiments carry forward to this challenging task. We use the network architecture specified in Mnih et al. (2015), and the hyper-parameters for TC as specified in Pohlen et al. (2018). Experimental details are specified in Appendix B. From the results, we see that HR-TD does not interfere with learning on the complex network, and does about as well as DDQN.

## 5.4 LINEAR FUNCTION APPROXIMATION

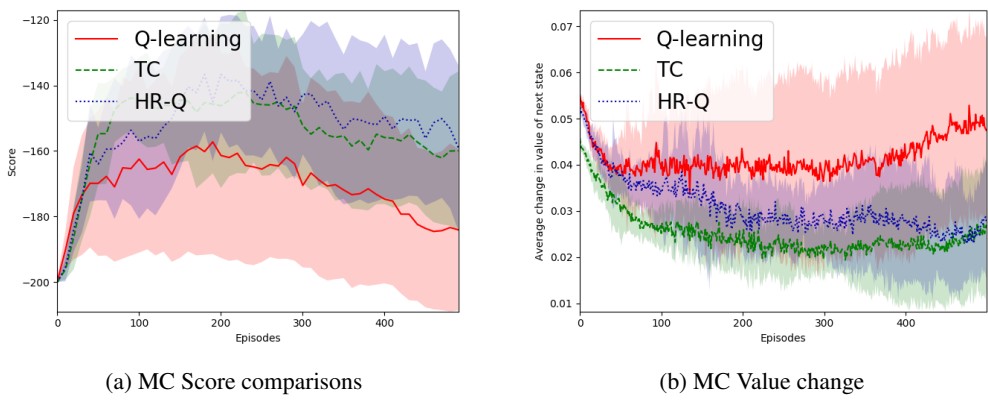

(a) MC Score comparisons       (b) MC Value change

Figure 4: Comparison on the Mountain Car domain.

Finally, we study HR-TD with the linear function approximation, we look at the Mountain Car domain. We expand the feature space using Fourier basis functions (Konidaris et al., 2011). All methods are trained with an order 6 Fourier basis expansion for Mountain Car Konidaris et al. (2011), which leads to 36 features for Mountain Car. We use a constant learning rate $\alpha = 0.01$ for all three methods. For HR-TD we initialize the regularization factor $\eta = 0.3$. Each episode is run

until we receive an episode termination signal from the Gym wrapper, which is a maximum of 200 steps if the goal is not reached. We show the learning curves for 1000 episodes, averaged over 20 independent runs. In Figure 4, we see that HR-Q and TC perform better than Q-learning. HR-Q also shows a more stable updates (changes value of next state less) than Q learning, and comparable to Q-learning with the added TC loss over the course of training.

# 6 CONCLUSION

In this paper, we analyze the problem of over-generalization in TD learning with function approximation. This analysis points to the potential pitfalls of over-generalization in TD-learning. Based on the analysis, we propose a novel regularization scheme based on the Hadamard product. We also show that with the right weight on the regularization, the solution of this method is the same as that of TD. Finally, we experimentally validate the effectiveness of our algorithm on benchmarks of varying complexity.

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

# Appendices

## A  PROBLEM WITH CTD: DOUBLE SAMPLING PROBLEM

Double sampling comes into effect whenever we need the product of two expectations. If an expression contains 3 expectations we will need three independent samples. Below we will first write out why residual gradients have a double sampling problem and why TD doesn't. Then we shall show why CTD has this problem, and might actually suffer from a triple sampling problem. Note that the double-sampling problem only exists in stochastic MDP problems. In a Deterministic MDP, double sampling will not be an issue.

$$\delta(s) = r(s, a, s') + \gamma V(s'|\theta) - V(s|\theta)$$
$$\mathcal{L} = \frac{1}{2}\mathbb{E}[\|\delta(s)\|]^2$$
$$\frac{\partial \mathcal{L}}{\partial \theta} = \mathbb{E}\left[\delta(s)(g(s) - g(s'))\right] \ldots \text{Residual Gradient}$$
$$= \mathbb{E}[\delta(s)]\mathbb{E}[g(s) - g(s')]$$
$$\frac{\partial \mathcal{L}}{\partial \theta} = \mathbb{E}\delta(s)g(s) \ldots \text{TD update}$$
$$\frac{\partial \mathcal{L}}{\partial \theta} = \mathbb{E}\left[\delta(s)g(s) - \frac{<g(s), g(s')>}{\|g(s')\|_2^2}g(s')\right] \ldots \text{Constrained TD update}$$

In the constrained TD update, the first term is the regular TD update, which has no double-sampling issues. However, the second term, $-\frac{<g(s),g(s')>}{\|g(s')\|_2^2}g(s')$, involves computing $s'$ in multi-places, and will need to sample multiple times to have an unbiased estimation, and thus have the double-sampling problems.

## B  EXPERIMENT DETAILS

### B.1  LINEAR FUNCTION APPROXIMATION

**Mountain Car**:
Basis Function: Fourier Basis, order 6
Max steps per episode: 200
Number of episodes: 500

### B.2  MLP-DQN

Layers: [64], Activation: ReLU, Optimizer: Adam
Replay Memory size: 50000
batch size: 32
minimum $\epsilon$ (for exploration) : 0.01
$\epsilon$ is decayed over $5\%$ of the episodes
$\eta$ is decayed as $\eta_t = \frac{\eta}{T+1}$, where $T$ is the episode number

| Technique | DQN | DQN+TC | HR-Q |
|---|---|---|---|
| Learning Rate | $10^{-2}$ | $10^{-2}$ | $10^{-2}$ |
| $\eta$ | - | - | 0.3 |

Table 2: Method specific hyper parameters for Mountain Car with linear FA

### B.2.1 MOUNTAIN CAR

Max steps per episode: 200, Number of episodes: 1000

| Technique | DQN | DQN+TC | HR-Q |
|---|---|---|---|
| Learning Rate | $10^{-3}$ | $10^{-4}$ | $10^{-3}$ |
| Target update | 500 | 500 | 500 |
| $\eta$ | - | - | 0.03 |

Table 3: Method specific hyper parameters for Mountain Car

### B.2.2 ACROBOT

Max steps per episode: 500, Number of episodes: 200

| Technique | DQN | DQN+TC | HR-Q |
|---|---|---|---|
| Learning Rate | $10^{-3}$ | $10^{-4}$ | $10^{-4}$ |
| Target update | 500 | 500 | 500 |
| $\eta$ | - | - | 0.01 |

Table 4: Method specific hyper parameters for Acrobot

### B.3 DQN FOR ATARI

Network: DQN architecture from Mnih et al. (2015), Optimizer: Adam
Replay Memory size: 100000, minimum $\epsilon$ : 0.01
$\epsilon$ is decayed over $5\%$ of the frames
Training Frames: 10M game frames, fed 4 at a time to network. (2.5 M agent steps)
$\eta$ is decayed as $\eta_t = \frac{\eta}{T+1}$, where $T$ is integer value of $\frac{t}{5000}$, and t is the iteration number.

| Technique | DQN | DQN+TC | HR-Q |
|---|---|---|---|
| Learning Rate | $10^{-4}$ | $5 \times 10^{-5}$ | $10^{-4}$ |
| Target update | 1000 | 2500 | 1000 |
| $\eta$ | - | - | $3 \times 10^{-2}$ |

Table 5: Method specific hyper parameters for Atari

## C POLICY EVALUATION LEARNING CURVES

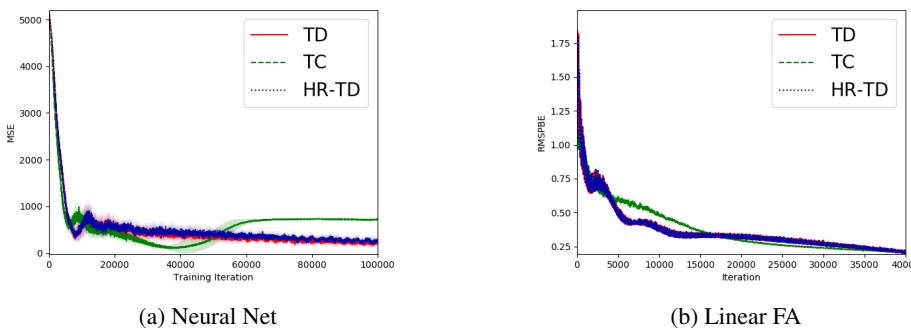

(a) Neural Net          (b) Linear FA

Figure 5: Comparison of policy evaluation on the Mountain Car domain.

