# OpenReview forum: "HR-TD: A Regularized TD Method to Avoid Over-Generalization"
_ICLR.cc/2019/Conference_

### Official Review · AnonReviewer3 · 2018-10-30
**Introduces a new variation on TD. Empirical results are not done well enough to support the claims of an improvement.**

**Rating:** 2
**Confidence:** 5

**Review:**

The paper introduces HR-TD, a variation of the TD(0) algorithm. The variant is meant to ameliorate a problem of ‘over-generalization’ with conventional TD. This problem is briefly characterized, but primarily it is presumed to be established by prior work. The algorithm is simple and a series of experiments are presented with it applied to Mountain Car, Acrobot, and Atari Pong, with both linear function approximation and neural networks (DDQN). It is claimed that the results establish HR-TD as an improvement over TD. However, I found the results unconvincing because they were statistically insufficient, methodologically flawed, and too poorly presented for me to be confident of the meaning of numbers reported. In addition, it is not hard to imagine very simple problems where the HR-TD technique would be counterproductive, and these cases were not included in the experimental testbeds.

The first weakness of the paper is with its characterization of the problem that it seeks to solve: over-generalization. This problem is never really characterized in this paper. It instead refers instead to two other papers, one published only in a symposium and the other with no publication venue identified.

The second weakness of the paper is the claim that it has done a theoretical analysis in Section 4.4. I don’t see how this section establishes anything of importance about the new method.

The problem with the main results, the empirical results, is that they do not come close to being persuasive. There are many problems, beginning with there simply not being clear. I read and reread the paragraphs in Section 5.1, but I cannot see a clear statement of what these numbers are. Whatever they are, to assess differences between them would require a statistical statement, and there is none given. Moreover to give such a statistical statement would require saying something about the spread of the results, such as the empirical variance, but none is given. And to say something about the variance one would need substantially more than 10 runs per algorithm. Finally, there is the essential issue of parameter settings. With just one number given for each algorithm, there are no results or no statement about what happens as the parameters are varied. Any one of these problems could render the results meaningless; together they surely are.

These problems become even greater in the larger problems.

A nice property of HR-TD is that it is simple. Based on that simplicity we can understand it as being similar to a bias toward small weights. Such a bias could be helpful on some problems, possibly on all of those considered here. In general it is not clear that such a bias is a good idea, and regular TD does not have it. Further, HR-TD does not do exactly a bias to small weights, but something more complicated. All of these things need to be teased apart in careful experiments. I recommend small simple ones.

How about a simple chain of states that are passed through reliably in sequence leading to a terminal state with a reward of 1000 (and all the other rewards 0). Suppose all the states have the same feature representation. If gamma=1, then all states have value 1000, and TD will easily learn and stick at this value even for large alpha, but HR-TD will have a large bias toward 0, and the values will converge to something significantly less than the true value of 1000.

That would be an interesting experiment to do. Also good would be to compare HR-TD to a standard bias toward small weights to see if that is sufficient to explain the performance differences.

---

### Official Review · AnonReviewer1 · 2018-11-03
**A new td method**

**Rating:** 3
**Confidence:** 4

**Review:**

This paper introduces a variation on temporal difference learning for the function approximation case that attempts to resolve the issue of over-generalization across temporally-successive states. The new approach is applied to both linear and non-linear function approximation, and for prediction and control problems. The algorithmic contribution is demonstrated with a suite of experiments in classic benchmark control domains (Mountain Car and Acrobot), and in Pong.

This paper should be rejected because (1) the algorithm is not well justified either by theory or practice, (2) the paper never clearly demonstrates the existence of problem they are trying to solve (nor differentiates it from the usual problem of generalizing well), (3) the experiments are difficult to understand, missing many details, and generally do not support a significant contribution, and (4) the paper is imprecise and unpolished.

Main argument

The paper does not do a great job of demonstrating that the problem it is trying to solve is a real thing. There is no experiment in this paper that clearly shows how this temporal generalization problem is different from the need to generalize well with function approximation. The paper points to references to establish the existence of the problem, but for example the Durugkar and Stone paper is a workshop paper and the conference version of that paper was rejected from ICLR 2018 and the reviewers highlighted serious issues with the paper—that is not work to build upon. Further the paper under review here claims this problem is most pressing in the non-linear case, but the analysis in section 4.1 is for the linear case.

The resultant algorithm does not seem well justified, and has a different fixed point than TD, but there is no discussion of this other than section 4.4, which does not make clear statements about the correctness of the algorithm or what it converges to. Can you provide a proof or any kind of evidence that the proposed approach is sound, or how it’s fixed point relates to TD?

The experiments do not provide convincing evidence of the correctness of the proposed approach or its utility compared to existing approaches. There are so many missing details it is difficult to draw many conclusions:
1) What was the policy used in exp1 for policy evaluation in MC?
2) Why Fourier basis features?
3) In MC with DQN how did you adjust the parameters and architecture for the MC task?
4) Was the reward in MC and Acrobot -1 per step or something else
5) How did you tune the parameters in the MC and Acrobot experiments?
6) Why so few runs in MC, none of the results presented are significant?
7) Why is the performance so bad in MC?
8) Did you evaluate online learning or do tests with the greedy policy?
9) How did you initialize the value functions and weights?
10) Why did you use experience replay for the linear experiments?
11) IN MC and Acrobot why only a one layer MLP?


Ignoring all that, the results are not convincing. Most of the results in the paper are not statistically significant. The policy evaluation results in MC show little difference to regular TD. The Pong results show DQN is actually better. This makes the reader wonder if the result with DQN on MC and Acrobot are only worse because you did not properly tune DQN for those domains, whereas the default DQN architecture is well tuned for Atari and that is why you method is competitive in the smaller domains.

The differences in the “average change in value plots” are very small if the rewards are -1 per step. Can you provide some context to understand the significance of this difference? In the last experiment linear FA and MC, the step-size is set equal for all methods—this is not a valid comparison. Your method may just work better with alpha = 0.1.


The paper has many imprecise parts, here are a few:
1) The definition of the value function would be approximate not equals unless you specify some properties of the function approximation architecture. Same for the Bellman equation
2) equation 1 of section 2.1 is neither an algorithm or a loss function
3) TD does not minimize the squared TD. Saying that is the objective function of TD learning in not true
4) end of section 2.1 says “It is computed as” but the following equation just gives a form for the partial derivative
5) equation 2, x is not bounded
6) You state TC-loss has an unclear solution property, I don’t know what that means and I don’t think your approach is well justified either
7) Section 4.1 assumes linear FA, but its implied up until paragraph 2 that it has not assumed linear
8) treatment of n_t in alg differs from appendix (t is no time episode number)
9) Your method has a n_t parameter that is adapted according to a schedule seemingly giving it an unfair advantage over DQN.
10) Over-claim not supported by the results: “we see that HR-TD is able to find a representation that is better at keeping the target value separate than TC is “. The results do not show this.
11) Section 4.4 does not seem to go anywhere or produce and tangible conclusions

Things to improve the paper that did not impact the score:
0) It’s hard to follow how the prox operator is used in the development of the alg, this could use some higher level explaination
1) Intro p2 is about bootstrapping, use that term and remove the equations
2) Its not clear why you are talking about stochastic vs deterministic in P3
3) Perhaps you should compare against a MC method in the experiments to demonstrate the problem with TD methods and generalization
4) Section 2: “can often be a regularization term” >> can or must be?
5) update law is a odd term
6)” tends to alleviate” >> odd phrase
7) section 4 should come before section 3
8) Alg 1 in not helpful because it just references an equation
9) section 4.4 is very confusing, I cannot follow the logic of the statements
10) Q learning >> Q-learning
11) Not sure what you mean with the last sentence of p2 section 5
12) where are the results for Acrobot linear function approximation
13) appendix Q-learning with linear FA is not DQN (table 2)

---

### Official Review · AnonReviewer2 · 2018-11-05
**Good to formulate the problem but issues in exposition and validation**

**Rating:** 4
**Confidence:** 4

**Review:**

The paper considers the problem of overgeneralization between adjacent states of the one-step temporal difference error, when using function approximation. The authors suggest an explicit regularization scheme based on the correlation between the respective features, which reduces to penalizing the Hadamard product.

The paper has some interesting ideas, and the problem is very relevant to deep RL. Having a more principled approach to target networks would be nice. I have some concerns though:
* The key motivation is not convincing. Our goal with representation learning for deep RL is to have meaningful generalization between similar states. The current work essentially tries to reduce this correlation for the sake of interim optimization benefits of the one-step update.
* The back and forth between fixed linear features and non-linear learned features needs to be polished. The analysis is usually given for the linear case, but in the deep setting the features are replaced with gradients. Also, the relationship with target networks, as well as multi-step updates (e.g. A3C) needs to be mentioned early, as these are the main ways of dealing with or bypassing the issue the authors are describing.
* The empirical validation is very weak -- two toy domains, and Pong, the easiest Atari game, so unfortunately there isn’t enough evidence to suggest that the approach would be impactful in practice.

Minor comments:
* there must be a max in the definition of v* somewhere
* V_pi is usually used for the true value function, rather than the estimate
* Sections 2.2 and 4.2 should be better bridged
* The relationship with the discount factor just before Section 5 is interesting, but quite hand-wavy -- the second term only concerns the diagonal elements, and the schedule on gamma would be replaced by a schedule on eta.

---

### Meta-Review · Area_Chair1 · 2018-12-10
**Issues with motivation and experiments**

**Confidence:** 5
**Recommendation:** Reject

**Metareview:**

All three reviewers raised the issues that (a) the problem tackled in the paper was insufficiently motivated, (b) the solution strategy was also not sufficiently motivated and (c) the experiments had serious methodological issues.